# Needs and Barriers of Teen Mothers in Rural Eastern Uganda: Stakeholders’ Perceptions Regarding Maternal/Child Nutrition and Health

**DOI:** 10.3390/ijerph15122776

**Published:** 2018-12-07

**Authors:** Josephine Nabugoomu, Gloria K Seruwagi, Kitty Corbett, Edward Kanyesigye, Susan Horton, Rhona Hanning

**Affiliations:** 1University of Waterloo. Ontario, Canada. 200 University Avenue West, Waterloo ON N2L 3G1, Canada; jnabugoomu@uwaterloo.ca (J.N.); kit.corbett@uwaterloo.ca (K.C.); sehorton@uwaterloo.ca (S.H.); 2Makerere University School of Public Health, Kampala. P. O. Box 7062, Kampala, Uganda; gseruwagi@musph.ac.ug; 3Uganda Christian University, Mukono. Bishop Road, P.O. Box 4, Mukono, Uganda; ekanyesigye@ucu.ac.ug

**Keywords:** Pregnancy in adolescence, infant, nutrition, health, needs, barriers, social cognitive theory, structural violence, gender

## Abstract

For adolescent mothers in rural Eastern Uganda, nutrition and health may be compromised by many factors. Identifying individual and environmental needs and barriers at local levels is important to inform community-based interventions. This qualitative study used interviews based on constructs from social cognitive theory. 101 adolescent mothers, family members, health-related personnel and community workers in Budondo sub-county (Jinja district), eastern Uganda were interviewed. Young mothers had needs, related to going back to school, home-based small businesses; social needs, care support and belonging to their families, employment, shelter, clothing, personal land and animals, medical care and delivery materials. Barriers to meeting their needs included: lack of skills in income generation and food preparation, harsh treatment, pregnancy and childcare costs, lack of academic qualifications, lack of adequate shelter and land, lack of foods to make complementary feeds for infants, insufficient access to medicines, tailored health care and appropriate communications. Using the social cognitive framework, this study identified myriad needs of young mothers and barriers to improving maternal/child nutrition and health. Adolescent-mother-and-child-friendly environments are needed at local levels while continuing to reduce broader socio-cultural and economic barriers to health equity. Findings may help direct future interventions for improved adolescent maternal/child nutrition and health.

## 1. Introduction

In Uganda, it was reported that 25% of adolescent girls (15–19 years) become pregnant, with this being more common in rural (27%) than urban areas (19%) [1]. In the Busoga region of Eastern Uganda, 21% of the adolescents aged 15–19 years have begun child bearing [1] placing them at risk of poor nutrition and health [2] and making it a significant public health concern. This study had a goal of understanding community stakeholder perceptions of the *needs* of teenage mothers in rural Eastern Uganda and also understanding the barriers they face in meeting those needs. In this case, the needs and associated barriers faced by young mothers and their infants in achieving adequate nutrition and health were conceptualized as are not only basic such as food, clothing and shelter but broad aspects of the well-being for both parties. The study applied the social cognitive theory (SCT) [3,4,5] to emphasize the individual and environmental (social, economic, physical, nutrition, health service) factors that interact to influence the behaviors of young mothers. Since the aim of this research was ultimately to guide community-level intervention, it was important to understand context from the perspectives of a range of stakeholders of adolescent maternal/child nutrition and health relevant to the geographic setting of the rural Jinja district.

## 2. Materials and Methods

This was a qualitative study using semi-structured face-to-face interviews with adolescent mothers and other stakeholders. The social cognitive theory (SCT) (Figure 1) was selected to understand how personal (or individual) factors and environmental factors interact reciprocally with behaviors [3,4,5,6] of adolescent mothers to influence their health and nutrition. The study, from an epistemological stance of postpositivism, sought to identify and understand determinants of the nutrition and health of adolescent mothers and their offspring, not limited to the research team’s prior knowledge or values [7,8,9,10,11,12,13]. The study therefore used both a deductive approach through a set of close ended questions reflecting constructs from the SCT [7,8,9,10,11,12,13] and an inductive approach through open ended research questions for freely given views of stakeholders [8,14].

### 2.1. Study Site

The study was conducted among rural communities of Budondo sub-county, Jinja district of Busoga region in Eastern Uganda [15]. A 2014 report found that Eastern Uganda has a poverty rate of 24.5% [16,17] making it the poorest of regions in the country [16]. The main occupation of the residents in Jinja is subsistence farming [15]. Budondo sub-county is located 25 km from Jinja Town, has a population of 51,560 (51.8% being females and 48.2% males) [15] and 36.3% of its residents are below the poverty line [18]. The sub-county has 6 government owned health centers [15].

### 2.2. Inclusion Criteria

All participants in the study had signed the study consent forms and had resided within Budondo sub-county or Jinja district for at least 3 years. In addition, adolescent mothers were aged 10–19 years, carrying their first pregnancy or with their first baby (0–12 months), and were attending or had attended school in Budondo sub-county at least 3 years prior to study.

### 2.3. Study Sample and Recruitment

Study participants who met the inclusion criteria were recruited through purposive sampling [8,19,20] by six community based Village Health Team members (VHTs) who served as study guides who assessed eligibility and invited eligible persons to participate in the study. Individual interviews were conducted with 101 individuals (Table 1) including 25 adolescent mothers plus family members, health personnel, community leaders and workers, and district administrators.

### 2.4. Data Collection

Interview guides, tailored to specific target study participants and translated into the Lusoga language, were used as data collection instruments for this study. Key questions were themed on individual factors and environmental factors (social, economic, physical, nutrition, and health service) relevant to needs and barriers of adolescent mothers. Interview guides were pre-tested in rural Butagaya sub-county with a few persons representative of the target groups. Interviews were conducted in privacy at the residences or work places of participants. Interviews were conducted from March to May 2016 by the researcher (JN) and took an average of 40 min. At the start of each interview, participants were welcomed by the researcher, told of the purpose of the interviews and assured of anonymity and confidentiality.

### 2.5. Data Analysis

Interview recordings were transcribed word for word then translated into English by a transcriber who was well versed in the Lusoga language. Codes were created from the transcribed interviews based on the constructs of the SCT model and *a priori* themes of needs and barriers under the categories of environmental factors and personal factors affecting adolescent mothers. Some inductively derived codes were added to label other information. Data management and analysis employed Atlas.ti 7.5.4 software; phrases in each transcript were linked to the created codes which were networked towards the major theme of adolescent maternal/child nutrition and health using thematic analysis [8,21,22,23] as shown in Figure 2. The Atlas.ti software which is manufactured by the ATLAS.ti Scientific Software Development was provided and uploaded by the Ugandan trainer who trained the researcher (JN) in qualitative data analysis in Uganda.

## 3. Results

### 3.1. Demographic Characteristics of the Respondents

Demographic characteristics of all study respondents are shown in the Table 1. Of the 101 study participants, over 65% were female. The family members in this study includes mothers and grandmothers of the young mothers. Married/cohabiting adolescent mothers and the infant’s fathers did not agree to participate in the study.

### 3.2. Perceived Needs and Barriers of Young Mothers

Perceived needs of and barriers for young mothers were reported by different stakeholders of adolescent maternal/child well-being and categorized according to individual and environmental (social, economic, physical, nutrition, health) levels (Figure 2).

### 3.3. Individual Level

#### 3.3.1. Needs at Individual Level

Adolescent mothers had a need for money to support their small businesses such as selling snacks; those businesses were continued by their parents at times when the young mothers are at school. Some of the young mothers made handcrafts like baskets, mats and ropes for sale, and all of them could cultivate food crops and rear animals with their parents. A need for modern medication and medical health care and to get back to school were also reported as individual needs of young mothers.

“*I so much wanted to get back to school because I have the heart to study, complete and also get a job. I need money for fees and so I make and sell spiral salty doughnuts for money needed. During school days I cook and leave the doughnuts with my mother and she sells them for me.*” Lactating Adolescent 1.

#### 3.3.2. Barriers at Individual Level

Over half of the stakeholders who commented reported that young mothers lacked knowledge in income generation skills, such as handcrafts. This is an individual factor that could be used to enhance their economic status. Some young mothers appeared to lack the confidence to take on new responsibilities of self-sustainability while others were perceived by stakeholders to be lazy at making handcrafts. In addition, over a third of the respondents who weren’t adolescent mothers, though that some young mothers stopped going to school on their own accord even when there were schools willing to keep them. Young mothers were also seen as lacking knowledge about practical food preparation to make food for themselves and their infants.

“*They [adolescent mothers] do not make handcrafts because they do not know how to make them. And this is not only in the villages but also in town schools.*” Agricultural Officer 1.

### 3.4. Social Environment Level

#### 3.4.1. Needs at Social Environment Level

According to half of the respondents who commented, young mothers in rural areas of Uganda needed to belong to their families. Respondents also reported that the mothers needed marriage as most of them were staying with their parents instead of their children’s fathers, many of whom were alleged to have denied responsibility or run off for fear of being arrested by police.

“*My parents do not trust me anymore. They abandoned and treated me badly, abusing and chasing me away. They [parents] lost hope in me, am looked at as being useless and a failure.*” Pregnant Adolescent 1.

Young mothers also expressed a need to belong to their families or those of the father of their baby as their families were sending them away yet their baby’s father or his parents were not taking them up. Young mothers identified the need to be loved, cared for and trusted by their families and community members instead of being abandoned or abused by them. This could have impacted their health as some young mothers thought of carrying out unsafe abortions. Stakeholders in the health service sector perceived that adolescent mothers were at a high risk of hypertension and depression during and after pregnancy due to stress at home.

“*Ever since my mother got to know that I am pregnant, she abuses me and wants to chase me away from home and she says that I should go to the person who made me pregnant yet he ran away. She does not show me love or care [about me] and never trusts me with anything, saying that I am useless.*” Pregnant Adolescent 2.

#### 3.4.2. Barriers at Social Environment Level

Some parents and community members were of the view that they had to be tough on the young mothers to teach them and other children in the family/community lessons not to repeat or make the same mistake. In this manner they appeared to justify the harsh treatment and abuse experienced by young mothers, which was reported by about half of the 70 respondents who commented.

“*When your daughter gets pregnant, you are not happy because you took her to school to study and look after you in future. Now instead of bringing home a qualification for a better future, one brings another burden, which is why we [parents] have to teach them [young mothers] a lesson. If you are not tough, even the young ones can make the same mistake. We [parents] are not happy about how these girls are embarrassing and letting us down.*” Mother.

Much as some study participants were of the view that schools could keep young mothers in school, some stakeholders gave a different view concerning especially pregnant adolescents. It was reported that some pregnant adolescents who might have wanted to continue with school were not accepted. In some cases those who went back after delivery were abused by their peers.

“*Girls may want to continue with school but they are not allowed to go to school while pregnant, they [school Administrators] say that they will spoil the rest of the girls…who may think that even when they get pregnant they will be allowed to continue with school. Those who deliver would have loved to go back to school but they fear to be abused by their peers and for others the parents believe that they are already spoilt and can’t study again.*” Grandmother.

### 3.5. Economic Environment Level

#### 3.5.1. Needs at Economic Environment Level

Young mothers in rural Uganda needed money to buy medicines, food, personal effects, and to pay for transport costs as their parents were too poor to support them. Jobs or self-employment were needed by young mothers so as to empower themselves economically.

“*I need a job to work so that I can provide for my needs and those of the baby because my mother does not have money to give me.*” Lactating Adolescent 2.

#### 3.5.2. Barriers at Economic Environment Level

Pursuit of various perceived avenues of improving the economic status of young mothers in rural Uganda were hindered by: the young women’s lack of academic qualifications for given job vacancies since they had dropped out of school; lack of financial support from their poor parents; lack of capital to become self-employed with small businesses; and their being pregnant or having a child being perceived by employers as linked to laziness or their productivity impacted by the need for time to rest or childcare. Income generation skills, e.g., agriculture and making handcrafts, would be an alternative to help in improving the economic status of young mothers, but the young mothers lacked markets for their handcraft products, lacked money to pay at facilities, like NGOs, that provided income generation training at a fee, or perceived distances to these NGOs to be too long. Moreover, an available government program of Operation Wealth Creation (formerly NAADS—National Agricultural Advisory Services) discriminated against young mothers through prioritizing and giving free seeds and animals for rearing to adults with established homes.

“*They (adolescent mothers) do not make handcrafts because they lack money to buy the materials needed as they no longer get them free of charge. Their parents also can’t help as they are very poor. Even what they make does not have market because they are not so attractive.*” Religious Leader 1.

“*NAADS [now known as Operation Wealth Creation] usually gives [agricultural items] to adults with established homes. And even if NAADS [Operation Wealth Creation] were giving youths seeds like beans, maize and animals to rear too, boys will overshadow the girls.*” Parish/Sub-county/District Administrator 1.

### 3.6. Physical Environment Level

#### 3.6.1. Needs at the Physical Environment Level

A house to sleep in and comfortable bedding for the young mothers and their infants were reported needs. In addition, soap and clothing, like maternity wear during pregnancy and shoes, were some of the physical needs expressed especially since they would feel unpresentable to health personnel and other community members. Perceptions of personnel in the health service sector indicated that in comparison to infants of adult mothers, infants of young mothers were more likely to have respiratory illnesses (flu, cough) and diarrhea that may result from the poor sleeping conditions.

“*Those pregnant girls need good dressing. They lack maternity dresses; for example, she may visit the hospital for ANC while putting on a T-shirt and sometimes they put on sandals made from old car tyres and sometimes they walk bare footed because they cannot even afford simple shoes. They do not have good bedding and their babies lack bedsheets and blankets.*” Health-related Personnel 1.

#### 3.6.2. Barriers at the Physical Environment Level

Long distances and slippery roads during rainy seasons presented a barrier that kept young mothers from accessing health centers or training programs. A third of the respondents reported that mothers lacked appropriate wear to visit health centers. Culture, as a social factor also presented barriers to meeting physical environment needs. For example, culturally, when a girl has sex or marries her parents may consider her as being a representative and the responsibility of the ‘in-laws’, and they may deny her adequate shelter.

“*They [young mothers] sleep under very poor conditions on nylon [sugar-empty] bags and papyrus mats and they usually sleep in the small huts of their brothers or in the ‘sitting room’ where it is cold because the parents take it that they are now in-laws.*” Educator.

Stakeholders also reported that owning of land by young mothers was made impossible by a culture that never allowed girls to inherit land. A lack of land and animals hindered young mothers from making decisions on what to grow or rear.

“*They [young mothers] don’t own land for themselves because our culture does not allow girls to inherit land. They also look after their parents’ cattle, goats and chickens; they do not have their own. They cannot decide on what to grow or rear; they follow what their parents want.*” Agricultural Officer 2.

### 3.7. Nutrition Environment Level

#### 3.7.1. Needs at Nutrition Environment Level

Young mothers in rural Uganda needed food of the right quality and quantity, as over half of the respondents who commented expressed that they only ate what was provided with no special consideration of their needs due to a lack of money for parents to buy the foods. It was perceived by health personnel that young mothers, in comparison to adult mothers, were more likely to have low maternal weight gain during pregnancy and anemia, and deliver infants with low birth weight due to this lack of food. There was also a need for the right complementary foods for the infants (6 to 12 months) and health personnel felt that infants of young mothers were also at a high risk of underweight.

“*They [infants] don’t get proper feeding like the right quantity of food or foods that they crave because their parents don’t have money to buy them all the foods that they need.*” Parish/Sub-county/District Administrator 2.

#### 3.7.2. Barriers at Nutrition Environment Level

Infants of some young mothers did not benefit from exclusive breastfeeding because their mothers lacked sufficient breast milk or had to go back to school or work. Exclusive breastfeeding was also, reportedly, hindered by young mothers’ beliefs and lack of knowledge or mentorship regarding breastfeeding, for example, an identified fear of breasts becoming wobbly on losing their firm shape (locally referred to as “socks”) or inability to manage discomfort, like sore nipples. Respondents also reported that as the mothers lacked appropriate foods and practical knowledge in food preparation, their infants were unlikely to receive the right complementary feeding.

“*Most of them (young mothers) feed the babies on what they eat, just mashing it. I have my neighbor who feeds the baby on cassava porridge yet the baby is very young but because she doesn’t have breast milk. It is recommended for the children to start complementary feeding when they are six months but some of them start it before it is six months and they give them hard foods like potatoes.*” Religious Leader 2.

“*Cookery practicals [for appropriate foods during and after pregnancy and complementary feeds] are not taught to young mothers yet that would help the mothers to be healthy. We only teach the foods to be eaten theoretically from charts. In addition, in Budondo Health Centre IV and Lukolo Health Centre III all mothers are sometimes taught how to cook using videos that are in English.*” Health-related Personnel 2.

### 3.8. Health Service Environment Level

#### 3.8.1. Needs at Health Service Environment Level

Adolescent mothers were in need of medicines and delivery materials like the “Mama Kit” (a pack of items to be used by mothers during delivery such as polythene to protect the bed, razors, cotton wool and gauze) but absence of these discouraged young mothers from seeking medical care or meant that those who went to health centers were not able to be served. Young mothers in rural Uganda reported a need for follow-up home visits by medical personnel after pregnancy and training in good newborn care practices, such as keeping a baby warm before and after bathing, and baby cord cleaning.

“*I am about to give birth to my baby. The midwife gave us a list of items to buy for delivery like mama kit, gloves, baby clothes, basin, soap but I do not have any of them. My mother cannot provide these, even getting money for buying medicines and food is a problem for her.*” Pregnant Adolescent 3.

#### 3.8.2. Barriers at Health Service Environment Level

Hindrances to seeking medical care that were identified included physical barriers, like long distances to the health centers and slippery roads during rainy seasons. Other barriers included: late reporting of medical personnel to the health centers; harsh treatment from the medical staff or midwives; long waiting lines; lack of maternity wear; and failure to obtain delivery materials e.g., Mama Kits, as some mothers lacked the money.

“*The medical workers reach at 10:00am finding long lines waiting. When they arrive, they first sign the attendance book and then converse and later work on us yet they have to leave early. They are so tough, they abuse us and don’t care much yet we will have come from far and waited for them.*” Pregnant Adolescent 4.

Respondents also reported that absence of medicines at the health center was a major barrier to seeking modern health services.

“*When there are no medicines and you go the first time they prescribe for you, [and a] second time and third time when there are no medicines, can you go back the fourth time?*” Health-related Personnel 3.

In addition, young mothers reported that results of all measurements regularly done for maternal weight, infant weights, malaria, HIV, anemia and blood pressure were told to mothers and recorded in their personal books, but were rarely communicated to them in a way that gave a clear explanation about their health status and that of their infants. Mothers took their weight and that of their infants without guidance from midwives. Midwives only waited to be told the figures for recording in the personal health record books of mothers. It was also perceived that young mothers lacked records on infant length and head circumferences. These gaps could hinder monitoring of the health of their infants. As well, a lack of designated space or time to address the specific health concerns of young mothers separate from adult mothers made it difficult to freely ask questions.

“*When the mothers go for ANC they are not separated from old mothers and these girls become uncomfortable. They cannot ask questions even if they have problems because they fear the adult mothers. Sometimes they [young mothers] are worked on last and they [midwives] abuse them asking them why they got pregnant while still young. This makes them fear to go back for ANC.*” Parish/Sub-county/District Administrator 3.

## 4. Discussion

Participants in this study identified that young mothers in rural Uganda hava many needs (personal, social, and environmental) that are not met by their families or the communities they lived in and hamper their nutrition, health and wellbeing. Our study agrees with earlier studies on needs of young Ugandan mothers such as: jobs and employment [24,25,26], food and shelter [27], knowledge in income generation skills, personal land and animals [26], and the need to overcome barriers e.g., abuse, stigma [26], lack of academic qualification [24,25,27], harsh treatment at health centers [25,26,27,28,29] and schools [26]. Sadly, these situations have not changed.

Key findings of this study include a need for support to meet even basic needs such as food and shelter, a sense of love and belonging to their families and those of the baby’s father, and medical care. The study noted that efforts to find solutions to such needs were also thwarted by a number of barriers at family, community or societal levels, including negative cultural beliefs, unaccommodating school environments, and lack of education and training, e.g., in income generation skill and newborn care. Other barriers included lack of medicines and staff, making health centers “empty”, lack of translation of health information to mothers, lack of home follow-ups, unfavorable environments at health centers, and lack of a designated space or time to address the specific health concerns of young mothers separate from adult mothers. These health-related barriers are in sharp contrast to the World Health Organization (WHO)’s six building blocks of an ideal healthcare system aimed at efficient and quality care including: service delivery, adequate workforce, information systems, accessibility to medicines, financing of the health sector, leadership and governance [30].

Pregnancy is experienced biologically by females, but some of the barriers faced relate to social responses to gender. The cultural beliefs that hindered young mothers from acquiring land or sharing houses with their parents represent a gender-based bias; favoring the boy child is a practice that is common in Uganda [31]. Moreover, harsh treatment from families, schools and the community to young mothers so as to ‘teach’ them or other female siblings/classmates lessons about premarital sex is also gender biased; the males that impregnated the girls were less severely affected, for example, those who were studying stayed in school [32]. Such acts of cultural and gender marginalization and inequalities suggest that young mothers in rural Uganda are victims of structural violence [33,34,35,36,37,38] that leads to social exclusion. These cultural gender biases that position adolescent mothers at the bottom of the social hierarchy are historically based processes and forces that cause further suffering [36,37,39]. While cultural transition takes time, participants pointed to some opportunities for local change.

The study design and analysis were based on the social cognitive theory. By framing the complex determinants of the wellbeing of young mothers across a range of individual and environmental levels, it helped to point to opportunities to overcome obstacles and support positive behavioral choices.

At an individual level, young mothers were found to lack education and skills. Opportunities for young mothers to better their lives might include giving them incentives to go back to school, and sensitizing schools and communities to support them. Seeking modern health care, going back to school, making money from home-base businesses, and cultivating crops and rearing animals are points of strength and resilience of young mothers that could be leveraged to improve their well-being. Some stakeholders’ perceptions that young mothers were lazy in making crafts or attending school may not have accounted for the demands of rearing infants or restrictions in some areas that prohibit pregnant or lactating mothers from attending school.

At a social level, barriers could be mitigated through fostering an attitudinal shift amongst those in positions of influences as well as service providers. For example, Kirstenstoebenau (2014) [40], with the help of the Forum for African Women Educationalists in Uganda, shared with district administrators, civil society organizations and policy makers the plight of adolescent mothers and other girls who needed to enroll back in school in West Nile region of Uganda after war and displacement. The audience was encouraged to standup for the well-being of not only girls but also young mothers who were out of school [40]. Such information sharing and advocacy could support a shift in accountability and attitudes among formerly abusive service providers towards young mothers. Sensitization and advocacy may also help support education and income generation, e.g., through parents faithfully caring for small businesses of young mothers who return to school (individual factor). It might also foster greater care for young mothers at a social level. Even if girls are not given land to own, sensitization could also be used to encourage all families to avail land for growth of food or rearing animals by young mothers since women are the main providers of agricultural labor in Africa [41,42]. A study in Ethiopia reported that a certified family ownership of land was demonstrated to be effective in helping women to own land and use it for economic growth [43], a case that could be borrowed for advocacy to help young mothers in rural Uganda to own land too.

Stigmatization and harsh treatment towards young mothers from their peers and educators in rural schools could be also be mitigated through sensitization of educators by those in positions of influence. The level of drop out in the primary schools is high with only 40% of Grade 1 girls completing primary circle with the main reason being teenage pregnancy [44]. This high rate of dropout may be a demotivating factor to returning to school by the adolescent mothers who have had a social and emotional disruption. Sensitization could also foster meeting basic needs for shelter, clothing, land (physical level); food for mothers and infants (nutrition level); and providing positive, accessible health care services (health service level). Over time, communities may transition in their acknowledgement of teen mothers as a collective responsibility.

At the economic level, avenues of partnering with organizations that would offer hands-on skill training need to be explored to support income generation by young mothers [45]. Support from such organizations is important because of the level of poverty in the rural setting of Uganda [16,17]; families, even if willing, may not be able to offer economic support to young mothers. Help from an external organization was demonstrated by the partnership with Wakisa ministries Uganda that helped young mothers enroll in a modern technology agricultural school, Agromax [46]. Similarly, the Teenage Mothers Project (TMP), funded by the Dutch organization Adopteer een Geit (Adopt a Goat) in Manafwa district (eastern Uganda), was reported to have improved the economic status of young mothers by giving each a goat and helping them enroll back into schools; the support of families, communities and a community-based organization, the African Rural Development Initiatives (ARDI), were all part of the TMP strategy of collective responsibility [24,47]. Other economic factors could be addressed at a macro or policy level. For example, policy might help overcome discriminatory practices, such as those mentioned previously of the organization (Operation Wealth Creation) that only gave seeds and animals to adults in established homes.

At the nutrition level, while lack of appropriate food and education about complementary feeds were barriers for infant nutrition, the practice of teaching theoretical content knowledge versus practical skills in healthy food preparation at health centers was not considered helpful. Skills-based nutrition and farming education [48,49,50,51,52,53] could support improvement in this area. Studies by Nabugoomu and colleagues (2015) showed that nutrition education of child caregivers in Uganda increased the adoption of orange-fleshed sweet potatoes to increase vitamin A intake by children (2–6 years), and had positive effects on caregiver knowledge, attitudes, and feeding practices [48,49].

At the level of the health service environment, lack of communication of health information in a way that was meaningful to young mothers is a barrier that could be addressed through health provider training, as has been observed elsewhere [54]. Improvement in abusive attitudes and inadequate care by service providers may need enforcement at the macro environment level of district and national government.

## 5. Conclusions

Young mothers in rural Uganda face tremendous barriers in meeting needs for adequate nutrition, health and wellbeing, including basic needs. These major needs include food and shelter; belonging and family care; knowledge, skills and resources for income generation; and medical care. In order to enhance their well-being and that of their offspring, collective efforts at individual, family, community and societal levels are needed. Guided by the social cognitive theory, the study provided a detailed depiction of needs and barriers, and also a glimpse of the strengths and opportunities that can lead to improvement. Findings of this study may help to direct future interventions for improvement of maternal/child nutrition and health.

## Figures and Tables

**Figure 1 ijerph-15-02776-f001:**
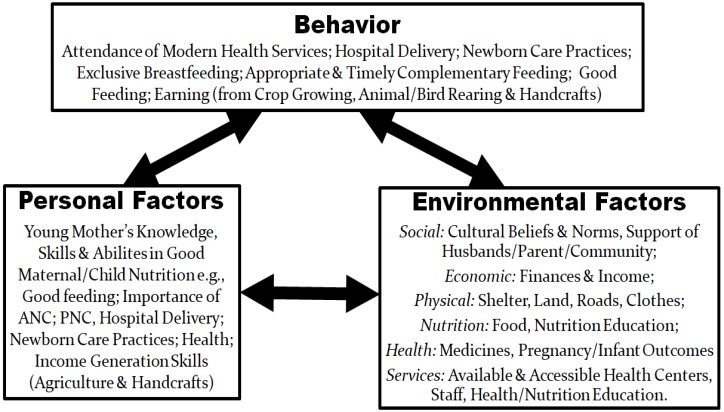
Social cognitive theory framework of perceived needs and barriers of adolescent maternal/child nutrition and health.

**Figure 2 ijerph-15-02776-f002:**
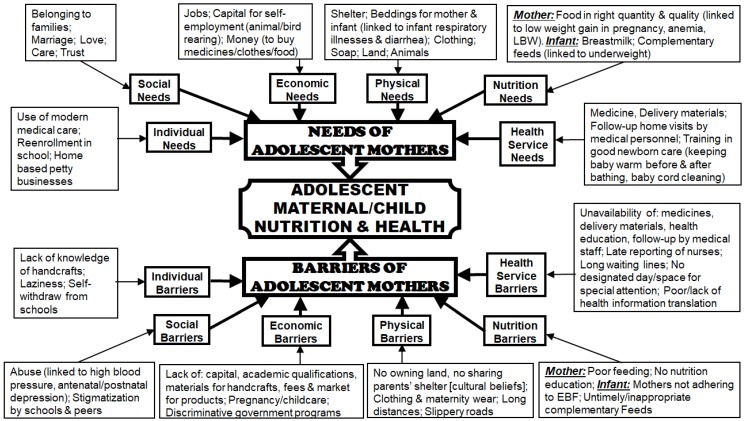
Thematic Network of Needs and Barriers of Adolescent Mothers as Perceived by Teenage Mothers and Stakeholders. *Adapted from:* [21].

**Table 1 ijerph-15-02776-t001:** Demographics of Study Respondents (*n* = 101).

Respondent Category	Gender		Number
	Male	Female	
Pregnant Adolescents	0	11	11
Lactating Adolescents	0	14	14
Mothers of Adolescent Mothers	0	5	5
Grandmothers of Adolescent Mothers	0	6	6
Educators	9	7	16
Health-related Personnel	4	15	19
Agricultural Officers	3	0	3
Religious Leaders	3	0	3
Parish/Sub-county/District Administrators	13	6	19
NGO Staff	3	2	5
Total	35	66	101

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
