# Peer review of "Needs and Barriers of Teen Mothers in Rural Eastern Uganda: Stakeholders’ Perceptions Regarding Maternal/Child Nutrition and Health"

_ijerph, 2018, doi:10.3390/ijerph15122776_

Round 1
Reviewer 1 Report
This is an important study which provides valuable insight into the significant health and social problems faced by pregnant and breastfeeding adolescents in a developing country. The qualitative data collected document the devastating barriers that young women face when they become pregnant and identifies the urgent need for services and supports. For this reason, I highly recommend that this manuscript be published with minor revisions, described below.
While you provide a good justification for the study (very high poverty rate, high teen pregnancy rate) in the Introduction, it is not clear from the title, abstract or introduction what they mean by ‘needs’ and ‘barriers’. When I later read the results, it became apparent that ‘needs’ include access to basic needs (food, clothing, shelter) as well as health care, education and family support. The barriers need to be explained/defined as well. E.g. barriers to optimal health care/nutritional health for her and her infant?
Materials and Methods:
On L 50, you state that you are interested in determinants of ‘nutrition and health’ of adolescent mothers. One would assume that these are behavioral or health indicators that you are attempting to explain. However, what you have listed in Fig 1 under “Behaviors” seems more feeding practices, access to health care and proxy measures of income. You also have some repetitiveness in the framework e.g. good feeding, hospital delivery and others in both ‘personal factors’ and in ‘behavior’. You have different names for the same concept e.g. Earning in behavior and income generation skills in “personal factors”. I recommend that the framework be revised so that it is clear and consistent.
It is not clear from the methods what you are actually assessing in the interviews. For example, you note that you use a deductive approach with a set of questions from the SCT and then some ‘open ended research questions’: please qualify if you had some closed ended and open ended questions, and the total number of questions. Please provide additional information on the data collection. E.g. Who conducted the interviews, were they in private, and how long did the interviews take?
While it may be acceptable given the thematic network approach that was used, I found it odd that you refer to Figure 2 under data analysis rather than the results section. The needs and barriers identified seem like they could provide a good overview of the results. It should be clear that this includes both perceptions of the teens themselves as well as other stakeholders. While the figure is already text heavy, some information that is omitted could lead to misinterpretations. For example, under ‘individual barriers’, you have ‘laziness’. This was a perception of stakeholder(s), not the teen mothers. Could you clarify in the title of the figure and the themes that it is
Results:
The results are well organized into sections describing findings pertaining to individual and environmental needs and barriers. Quotes are relevant and illustrate the themes identified. I did find some inconsistencies which should be addressed in the discussion: the negative perceptions of stakeholders concerning individual barriers (L 119-127) (e.g. that girls are lazy and stopped going to school even when schools would keep them) is incongruent with the grandmother’s quote on L 160-164, which indicates that girls are not allowed to go to school by administrators and parents.
Discussion:
Generally, key issues are addressed in the discussion.
L 288-293: Again, explain what you mean by ‘needs’ (L: 288). You found that food for the young mothers and their infants was inadequate- did no studies report that?
L 313-315: After describing this desperate situation, can you introduce some ideas as to what is needed to move forward and rectify it? You do discuss this later but ideas need to be linked.
L 316: This paragraph begins by stating that the study is based on SCT. However, the rest of the paragraph is not consistent with this sentence. I would expect a discussion here of how appropriate/effective it was to use the SCT in this context.
L 321- 323: Given that they are being abused, have no support and inadequate food, and no land ownership or ability to make decisions on their family farm, how realistic is it to say that they will be able to improve their well being by cultivating crops, rearing animals or make money from a home based business? Leveraged by whom? In the next paragraph (L 324-336), you do a good job explaining how some of these barriers can be addressed- perhaps you could combine this earlier statement and this paragraph.
L 337-370: I found that the discussion focused more on solutions to address the problem rather than discussing the findings in the context of other literature.
Conclusions:
It is important to ensure that your conclusions reflect your findings; I would prefer to see more detail concerning key themes.
Some additional minor edits:
L 55: wording- ‘questions for freely given views’: would be clearer if you said ‘ questions designed to obtain views…”
L 74: Recommend that this sentence be first in the paragraph to improve flow.
L 87: please include who transcribed the recordings.
L 99: are shown in Table 1.
L 252: explain ‘Mama kit”.
L 367-370: references needed
Author Response
Thank you for your thorough review and helpful feedback.
1. The Introduction will be modified to make the meaning of ‘needs and barriers’ clearer, as follows:
Line 38-42: [1“Line 38-42: [1] placing them at risk for poor nutrition and health [2]. This study had a goal of understanding community stakeholder perceptions of the needs of teenage mothers in rural Eastern Uganda and also understanding the barriers they face in meeting those needs. In this case, the needs and associated barriers faced by young mothers and their infants in achieving adequate nutrition and health were conceptualized as not only basics such as food, clothing and shelter but broad aspects of well-being for both mothers and infants.
2. Materials and Methods: We have attempted to make the use of words more consistent and clarify the Figure 1 to better differentiate between Individual capacity (knowledge, skills and abilities) and behavioral practices.
3. Yes, there were close ended questions too. The sentence was improved to…. “a deductive approach through a set of close ended questions reflecting constructs from the SCT [7,8,9,10,11,12,13] and an inductive approach through open ended research questions for freely given views of stakeholders [8,14].
4. Data collection has been edited to include, “Interviews were conducted in privacy at the residences or work places of participants. Interviews were conducted by the researcher (JN) and took an average of 40 minutes.”
5. Fig 2 does fit with both the description of the analysis (and the results. (The thematic network is a summary of what came out of linking conversation phrases to codes.) We therefore refer to it in the data analysis section and now also in the Results section 3.2 (14 lines later). The title has been changed to “Thematic Network of Needs and Barriers of Adolescent Mothers as Perceived by Teenage Mothers and Stakeholders.”
6. On barriers at individual level, edits are “… while others were perceived by some stakeholders to be lazy at making handcrafts. In addition, over a third of the respondents who weren’t adolescent mothers, thought that some young mothers stopped going to school on their own accord even when there were schools willing to keep them.” It has been also clarified in the discussion as “Some stakeholders’ perception that young mothers were lazy in making crafts or attending school may not account for the demands of rearing infants or restrictions in some areas that prohibit pregnant or lactating mothers from attending school.”
7. For the discussion, we clarifiedneeds to align with the rephrased objective:” Participants in this study identified that young mothers in rural Uganda have many needs (personal, social, economic and environmental) that are not met by their families or the communities they lived in and hamper their nutrition, health and wellbeing”. Thanks for pointing out the oversight of the reference to food insecurity. This has been added to the next sentence as follows: “Our study agrees with earlier studies on needs of young Ugandan mothers such as: jobs and employment [24,25,26], food and shelter [27], knowledge in income generation skills, personal land and animals [26], and the need to overcome barriers e.g., abuse, stigma [26], lack of academic qualification [24,25,27], harsh treatment at health centers [25,26,27,28,29] and schools [26].”
8. As a brief link to the later ideas, we added the phrase. “While cultural transition takes time, participants pointed to some opportunities for local change.”
9. Re. the comment (line 316), a few tweaks were made to help frame the SCT better. The paragraph now reads: “The study design and analysis were based on the social cognitive theory. By framing the complex determinants of the wellbeing of young mothers across a range of individual and environmental levels, it helped to point to opportunities to overcome obstacles and support positive behavioral choices. “
10. On owning land and cultivation, this suggestion has been improved as, “Even if girls are not given land to own, sensitization could also be used to encourage all families to avail land for growth of food or rearing animals by young mothers since women are the main providers of agricultural labor in Africa [44,45]. A study in Ethiopia reported that a certified family ownership of land was demonstrated to be effective in helping women to own land and use it for economic growth [43], a case that could be borrowed for advocacy to help young mothers in rural Uganda own land too.”
11. Findings of the study are discussed in relation to other studies in paragraph one of the discussion and as relevant when suggestions of solutions were drawn from earlier studies.
Conclusions:
12. The conclusion now reiterates key themes: “Young mothers in rural Uganda face tremendous barriers in meeting needs for adequate nutrition, health and wellbeing, including basic needs. These major needs include food and shelter; belonging and family care; knowledge, skills and resources for income generation; and medical care.”.
13. Rewording to “questions designed to obtain views”, done.
14. This sentence moved to start of paragraph to maintain flow. “Study participants who met the inclusion criteria were recruited through purposive sampling [8,19,20] by six community-based Village Health Team members (VHTs) who served as study guides who assessed eligibility and invited eligible persons to participate in the study.”
14. Transcriber included as, “Interview recordings were transcribed word for word then translated into English by a transcriber who was well versed in the Lusoga language.”
15. Word “shown” in Table 1 added
16. Explanation of Mama kit added as, “A “Mama Kit” is a sealed package of delivery materials including plastic/polythene sheet, cord ties, razor blades, cotton wool with gauze, gloves and a child health card”
17. A general reference to health literacy has been added
Reviewer 2 Report
Summary
This paper presents results of a qualitative study using social cognitive theory to identify barriers and needs faced by adolescent mothers in rural Uganda in terms of the health and nutrition of themselves and their infants. Through interviews with pregnant and lactating adolescent mothers as well as the mothers of the adolescent mothers and other key community members, the authors identify a number of needs and barriers at the individual, family, community, and societal levels that contribute to adolescent mothers not having adequate health and nutrition for themselves and their infants.
This is an important topic, and I commend the authors for taking it on. Below, I provide some broad comments for consideration that may help improve the impact and readability of the paper as well as some specific comments.
Broad comments
The introduction needs to be expanded. The intro should include more/better motivation for your study within the specific context in which it took place (for example, can you present some evidence from the literature and/or statistics showing shortfalls in the health and nutrition of adolescent mothers and their infants in rural Uganda?). The intro should also more explicitly describe how the results of your research might be used to improve the health and nutrition situation for these girls.
In terms of organization of the results, I think it would be more insightful to present each set of needs and barriers according to the responses of the adolescent girls and then the responses of other respondents. You would not need to make individual subsections, but perhaps separate paragraphs in which it is very clear whose responses are being summarized/reported.
Specific comments
Line 36: I’m assuming the 25% is referring to adolescent girls, not all adolescents, correct? If so, revise to, “…adolescent girls (15-19 years)…” Ditto for line 38
Lines 38-39: For clarity, revise to, “…making it a significant public health concern.”
Line 42: Why are most of the factors listed in parentheses separated by “/” rather than commas?
Figure 1: Wouldn’t “newborn care practices” be a behavioral factor, not a personal factor? What is meant by “good feeding”? Also, please define acronyms used in the figure.
Line 64: residents, not residence.
Line 70: Why was school attendance an inclusion criteria? What about the perceptions of adolescent mothers who did not attend school?
Section 2.4: Please indicate when the data were collected (date range). Also, were interviews conducted in private? I think this might be particularly important for the interviews conducted with the adolescent mothers to ensure they felt comfortable being open and honest.
Line 113: Not clear what is meant by “biomedical” health care. Please define or revise.
Line 121: “Some appeared to lack the confidence to take on new responsibilities of self-sustainability…” Is this an opinion of the study respondents or of the interviewers? It is not clear from the way it is written. Moreover, I’m not sure that expecting these very young adolescent girls with new babies to become “self-sustainable” is a reasonable (or helpful to promote health and nutrition) expectation. Rather, it seems that what they need is a lot of support: financial, emotional, etc.
Line 228: Complementary feeding is usually defined to cover the period from 6 months onward (see https://www.who.int/nutrition/topics/complementary_feeding/en/).
Line 252: Please describe what is included in the “Mama kit.”
Line 311: Using the phrase “less severely affected” implies that boys were affected. In what ways are the boys affected? Did effects on the boys come up in the interviews?
Line 336: Effective at achieving what? Please specify and situate the findings into your specific context (i.e. adolescent mothers).
Author Response
Thank you for your helpful feedback. The manuscript has been edited to respond to your suggestions. Please note that the original text is in black and the proposed revisions in response to the reviewer’s comments in red.
1. While no specific statistics related to nutrition and health for adolescent mothers and their infants in rural Uganda are documented, the introduction now cites general concerns: “Line 38-42: [1] placing them at risk for poor nutrition and health [2] and making it a significant public health concern. The introduction has now been expanded and we hope better captures the motivation and aim.
2. Responses from adolescent mothers and other stakeholders have been well labelled.
3. “girls” added to adolescent.
4. Revision of “making it a significant public health concern.” done.
5. Factors separated by commas
6. The individual factors encompass knowledge and attitudes, such as and knowledge of new born care and the behavioral factor the practices. This is clearer in the revised figure.
7. Acronyms have been removed. These would have been defined as “(ANC = antenatal care; PNC = postnatal care)”.
8. Residence changed to residents.
9. School attendance was at least three years prior was an important inclusion criteria to capture information concerning experiences of adolescent mothers who stay or dropout of school and help identify whether schools might be perceived as potential points of intervention.
10. Section on data collection edited to begin as “Interviews were conducted from March to May 2016 by the researcher (JN) and took an average of 40 minutes.”
11. bio-medical changed to “medical”.
12. On barriers at individual level, edits are “Some young mothers appeared to lack the confidence to take on new responsibilities of self-sustainability while others were perceived by stakeholders to be lazy at making handcrafts. In addition, over a third of the respondents who weren’t adolescent mothers, though that some young mothers stopped going to school on their own accord even when there were schools willing to keep them.”
It has been also clarified in the discussion as “Some stakeholders’ perceptions that young mothers were lazy in making crafts or attending school may not have accounted for the demands of rearing infants or restrictions in some areas that prohibit pregnant or lactating mothers from attending school.”
13. 7—12 months changed to “6-12 months” to avoid confusion. It is true that complementary feeding covers the period of 6-12 months but we grouped young mothers in pregnant, lactating (0-6months) and lactating 07-12 months (technically >6 months). The 0-6 months group was to help capture information on EBF.
14. Explanation of Mama kit added as “A “Mama Kit” is a sealed package of delivery materials including plastic/polythene sheet, cord ties, razor blades, cotton wool with gauze, gloves and a child health card”
15. Not many effects came up for the boys. Only one came up where boys implicated escaped from the homes or were taken to prison if caught. Otherwise, most of them continued with education within or away from their villages. The discussion to say “the males that impregnated the girls were less severely affected, for example, those who were studying stayed in school” has been referenced.
14. Sentence edited to “effective in helping women to own land and use it for economic growth [43], a case that could be borrowed for advocacy to help young mothers in rural Uganda own land too”.
Reviewer 3 Report
Your introduction of specific needs and barriers does not come until the discussion section. It seems the specifics ought to be brought into focus for the reader in the introduction.
I believe the barrier of uncaring or abusive service providers merits more than one sentence in the discussion -- you are talking about a BIG barrier and perhaps even culture change -- not something to gloss over.
The writing needs a good editing -- lines 60 to 76 seem stilted and wordy -- you don't need to list the percent of males, for example, when you provide percent of females.
Does stress and abuse at home cause hypertension? This needs support and citation if possible -- there are many reasons why a young pregnant woman might be hypertensive.
Author Response
Thank you for your helpful feedback. Please note that the original text is in black and the proposed revisions in response to the reviewer’s comments in red.
1. The introduction has been improved to better describe ‘needs’ and ‘barriers’.
“In Uganda, it was reported that 25% of adolescent girls (15-19 years) become pregnant, with this being more common in rural (27%) than urban areas (19%) [1]. In the Busoga region of Eastern Uganda, 21% of the adolescents aged 15-19 years have begun child bearing [1] placing them at risk of poor nutrition and health [2] and making it a significant public health concern. This study had a goal of understanding community stakeholder perceptions of the needs of teenage mothers in rural Eastern Uganda and also understanding the barriers they face in meeting those needs. In this case, the needs and associated barriers faced by young mothers and their infants in achieving adequate nutrition and health were conceptualized as are not only basic such as food, clothing and shelter but broad aspects of the well-being for both parties. The study applied the social cognitive theory (SCT) [3,4,5] to emphasize the individual and environmental (social, economic, physical, nutrition, health service) factors that interact to influence the behaviors of young mothers. Since the aim of this research was ultimately to guide community-level intervention, it was important to understand context from the perspectives of a range of stakeholders of adolescent maternal/child nutrition and health relevant to the geographic setting of the rural Jinja district.”
2. On abusive service providers, the of social factors has been added to read as “Such information sharing [referring to reference 39] and advocacy could support a shift in accountability and attitudes among formerly abusive service providers towards young mothers.” “It is acknowledged at the end of the discussion, “Improvement in abusive attitudes and inadequate care by service providers may need enforcement at the macro environment level of district and national government.”
3. Some editing was done on sections relating to study site and inclusion criteria. This was also noted by other reviewers.
4. The issue of stress and hypertension was a perception of stakeholders in the health service sector, as stated in the manuscript We have not verified other perceptions in this manuscript, though this one seems feasible (e.g., Clayton J. Hilmert, PhD, Christine Dunkel Schetter, PhD, Tyan Parker Dominguez, PhD, MPH, MSW, Cleopatra Abdou, MA, Calvin J. Hobel, MD, Laura Glynn, PhD, and Curt Sandman, PhD Stress and Blood Pressure During Pregnancy: Racial Differences and Associations With Birthweight Psychosom Med. 2008 Jan; 70(1): 57–64. PMID: 18158373
OR Yunxian Yu, PhD,1 Shanchun Zhang, PhD,1 Eric B Mallow, MD,2 Guoying Wang, MD,2 Xiumei Hong, PhD,2 Sheila O. Walker, PhD,2 Colleen Pearson, BA,3 Linda Heffner, MD,4 Barry Zuckerman, MD,3 and Xiaobin Wang, MD2 The Combined Association of Psychosocial Stress and Chronic Hypertension with Preeclampsia. Am J Obstet Gynecol. 2013 Nov; 209(5): 10.1016/j.ajog.2013.07.003. PMID: 23850528